# Development and Validation of a Mobile Phone Application Developed for Measuring Dietary Fiber Intake

**DOI:** 10.3390/nu13072133

**Published:** 2021-06-22

**Authors:** Rebecca Ahlin, Ida Sigvardsson, Viktor Skokic, Rikard Landberg, Gunnar Steineck, Maria Hedelin

**Affiliations:** 1Department of Oncology, Sahlgrenska Academy, University of Gothenburg, 40530 Gothenburg, Sweden; ida.sigvardsson@vgregion.se (I.S.); viktor.skokic@gu.se (V.S.); gunnar.steineck@oncology.gu.se (G.S.); maria.hedelin@oncology.gu.se (M.H.); 2Department of Molecular Medicine and Surgery, Karolinska Institutet, 17176 Stockholm, Sweden; 3Department of Pelvic Cancer, Karolinska University Hospital, 17176 Stockholm, Sweden; 4Department of Biology and Biological Engineering, Chalmers University of Technology, 41296 Gothenburg, Sweden; rikard.landberg@chalmers.se; 5Regional Cancer Center West, Sahlgrenska University Hospital, 41345 Gothenburg, Sweden

**Keywords:** validation, mobile phone application, dietary record, dietary fiber, nutrition

## Abstract

We have developed a mobile phone application for measuring the intake of dietary fiber and validated the ability of the application to accurately capture this intake against measurements registered by a dietary record. We also investigated what food groups contributed most to the total, soluble, and insoluble dietary fiber intake. Twenty-six randomly selected Swedish women aged 35–85 years were included and randomized to either start to register dietary intake in the application or by a dietary record, during three consecutive days. After a washout period of at least two weeks, the participants used the other method. We found that the difference in measured mean fiber intake between the dietary record and the application was two grams independent of the total intake per day. A statistically significant correlation between fiber intake as measured by the two methods was found (rho = 0.65, *p* < 0.001). Vegetables and roots were the predominantly contributing foods to total and soluble fiber intake. Bread and crackers contributed most to insoluble fiber intake. In conclusion, the application may be considered as a useful and easy-to-use method to measure dietary fiber intake.

## 1. Introduction

To capture habitual dietary intake is challenging because dietary assessment methods have several limitations [1]. For validation of methods intended to measure nutrient intakes, a weighed dietary record is often used as the reference method [2,3,4]. Weighed dietary records have the potential to provide accurate information on amounts and types of foods consumed over a period [5]. Registering a dietary record during a seven or more days-long period may create a risk of negligence because recording the same kind of information daily can become more and more tiring with each additional day of observation, imposing great demands on the person keeping the record [6]. As the technical society develops and almost everyone owns a mobile phone, there is an opportunity to use mobile phone applications (hereafter-called application) in research. Applications can be used to monitor compliance in intervention studies and may be used to improve the feasibility and accuracy of assessing dietary intake [7]. Compared to other more conventional methods of dietary assessment, an application has the advantage of being portable and easily accessible for participants, and individual nutrition data can be directly transferred to facilitate making analyses. However, there are only a few studies that have compared the performance of different applications and their validity to estimate whole diets or specific food components such as dietary fiber, compared to dietary records [7].

The European Safety Authority (EFSA) defines dietary fiber as non-digestible carbohydrates plus lignin [8]. The definition has been debated for years and it is still up to international authorities to decide if oligosaccharides should be included or not [9]. The most common dietary fiber sources in the Western world are whole-grain cereal products, vegetables, fruits and berries, legumes, seeds, and nuts [10]. Dietary fiber can be divided into water-soluble- and water-insoluble fibers (e.g., wheat bran) [11]. Soluble fiber can further be divided into viscous or gel-forming (e.g., psyllium husk) and non-viscous (e.g., inulin). Dietary fiber has several health benefits related to its properties. Soluble fiber—especially non-viscous fiber—are fermented by bacteria in the colon and form short-chain fatty acids [11,12]. The short-chain fatty acid butyrate is especially important for colonic function [12]. 

In the ongoing FIDURA-study (https://clinicaltrials.gov/ct2/show/NCT04534075?term=fidura&draw=2&rank=1; accessed date 21 June 2021), we have developed an application to measure the intake of dietary fiber among patients during pelvic radiotherapy. An application could provide a more user-friendly, convenient, and acceptable method than other more time-consuming and demanding existing dietary assessment methods. 

Thus, we wanted to investigate the performance of the application to measure the intake of dietary fiber accurately and conveniently in a population-based study among Swedish women aged 35–85 years. Our goal was to develop an application that will capture the mean dietary fiber intake among users and to validate it against an often-used reference method, i.e., a three-day weighed dietary record. Secondarily, we wanted to examine which food groups contributed most to the total, soluble- and insoluble dietary fiber intake among the participants.

## 2. Materials and Methods

### 2.1. Study Population

A randomly selected group of women (*n* = 125) was identified from the Swedish Population Register and each received an invitation letter via mail. The inclusion criteria were women, aged 35–85 years living in the Region Västra Götaland, Sweden. Only women were included as the pre-study of FIDURA only includes women. The exclusion criteria were not being able to understand written Swedish, not having access to a mobile phone or tablet, or lacking the ability to download applications. Phone numbers were missing for 37 of the 125 women, thus these were sent an alternative letter and were asked to themselves contact the study office for more information if interested in participating. We got in contact with and gave oral information about participation to 70 of the 88 women with phone numbers available. Two of the women with unavailable numbers contacted us themselves. Thirty-five women orally agreed to participate in the study and received study material by mail. The main reasons for declining study participation were lack of interest or lack of time (*n* = 12) and difficulties with using applications (*n* = 5). Seven women dropped out before they signed the informed consent document, and the reasons were mainly related to their own or family illness. A description of the selection procedure is presented in Figure 1. Twenty-eight women signed the informed consent sent by mail. One woman dropped out during the study due to problems with using the application. Participants were recruited from April to June 2020 and the last participant finished the second registration in November 2020.

### 2.2. Study Design

The study procedure is presented in Figure 2. Participants were randomized to either start recording the three-day dietary record or registering dietary intake in the application for three consecutive days. Randomization was executed by the study secretariat using a closed envelope in blocks of four (two to start with the application plus two to start with the dietary record), to achieve a similar number of participants in both groups. Participants received study material by mail, and instructions of both assessment methods included registration from Wednesday to Friday. Participants reported special diets and allergies at the inclusion and self-reported weight and height in connection with the registration of the first assessment method. The first registration was followed by a washout period of at least two weeks before completing the other assessment method. The average time of the washout period (*n* = 27) was 3.1 weeks (SD ± 2.1; median = 3.0; range = 1–11 weeks). Printed instructions for the second assessment were sent out after the completion of the first assessment. A portion guide [13] was sent together with the three-day dietary record. The participants were instructed to eat and drink as usual and not modify their dietary intake during the recordings. No physical meetings were held; all contact was made by phone and all tasks were handled by the study secretariat.

After completion of both dietary assessment methods, all participants were contacted for follow-up questions concerning the use of the application and the dietary record. Questions included if they had missed any food items in the application and the question “What did you think about using the application versus using the dietary record?”. We asked if they were interested in feedback on the results of their recordings, including information about fiber recommendations in Sweden. Due to the ongoing pandemic Covid-19, participants were also asked if they thought the pandemic situation had any effect on their food intake between the two registration periods. The study was approved by the Swedish Ethical Review Authority (Dnr 2019-06252).

### 2.3. Mobile Phone Application

The application was developed for the FIDURA-study in 2019. The application includes the headlines “daily dietary intake”, “daily stools”, “weekly events” and “my intake”. “Daily dietary intake” includes 137 common high-fiber foods and equivalents containing a low amount or no dietary fiber (e.g., whole grain rice and white rice) and 51 dishes from recipes. The food items are searchable but can also be selected under the categories “bread/crackers”, “fruits/berries”, “vegetables/roots/potatoes”, “legumes”, “nuts/seeds”, “grains/cereals/flour”, “pasta/rice/food grains”, “snacks”, “plant-based substitutes” (Quorn and soy protein products), “saved meals”, and “recipes”. Users of the application register specific quantities of food items in numbers, slices, deciliters, or tablespoons to one decimal place. The recipes are registered in portions and are easy-to-prepare, typically prevalent in the Swedish diet (e.g., sandwiches, porridges, stews, soups), and could be tolerated during pelvic radiotherapy. The recipes were added to the application to ease the registration for participants in the FIDURA-study. “My intake” shows users’ registered foods and total intake of dietary fiber per day for all registered days. Added food items can be edited and deleted and new items can be added. 

Participants were instructed to only use “daily dietary intake” and “my intake”. Instructions included registering food items containing dietary fiber in the existing categories except for “recipes”; participants would instead select each food item in the dish from the other categories. If they did not find the right food item, they were instructed to choose the most similar item or contact the study secretariat if they were unsure. 

The total fiber amounts of the food items are based on information from the Swedish Food Agency food database (version 2020-01-16) [14]. The proportion of soluble and insoluble fiber is based on comparable food items from the Finnish Food Composition Database Fineli release 19 (3 March 2018) [15] because the Swedish database is lacking this information. 

### 2.4. Dietary Record

For the dietary record, participants were instructed to note all consumed foods and drinks—except tap water—and the quantities of them. Instructions included primarily using a household scale to weigh foods and drinks or otherwise household measures or the portion guide they had received. Participants were also asked to record information such as cooking method, percentage of fat, and brand of the foods and drinks consumed. Dietist Net (version 2020-01-23) [16]—based on the Swedish Food Agency food database (last updated 2020-01-16)—was used to calculate nutritional intake registered from the dietary records.

### 2.5. Definition of the Most Common Fiber Sources

Food items in the dietary records were categorized into the same categories as in the application. If an appropriate category was missing for food items contributing to fiber intake in the dietary records, they were defined as the food group “other” and included for example flavored dairy products, cookies, and sweets. Meals reported in the dietary records that were registered as a pre-calculated meal instead of their main ingredients in Dietist Net were defined as the food group of the item with the assumed greatest contribution to dietary fiber in the meal.

### 2.6. Complimentary Food Items in the Mobile Phone Application

Food items in the “other” category from the dietary record and items that participants reported they missed in the application in this study and the preparatory study of FIDURA were considered as potential items to add to the application.

### 2.7. Statistical Analysis

The age of enrolled women was set to their age at the date of signed consent. For non-participants, the age was set to their age at first contact by phone. For women not reachable or who had no available phone number, the age was set to their age at the time when the invitation letter was sent out. Individual mean nutrition intakes were calculated as the sum of all observed values divided by the three days. The normal distribution of dietary fiber intakes was analyzed by using the Shapiro-Wilks test. Mann-Whitney U test was used to compare mean intakes between the two randomized groups since fiber data were non-normally distributed.

Validation of the application was conducted by comparing calculated fiber intakes measured by the application and the dietary record, using the Pearson correlation coefficient. A Bland-Altman plot was used to visually illustrate the agreement of dietary fiber intakes as indicated by data collected by the application and the dietary record. The difference between the results from the two methods was plotted against the mean reported fiber intake of the two methods. A linear regression analysis regressing the difference between the results from the two methods on the mean measurements of the two methods was performed to investigate the presence of a trend in the measurement deviations as a function of the mean measurements.

To evaluate if demographic characteristics affected the relative deviation in terms of mean fiber intake, regression analyses were used. Age, mean reported dietary fiber intake of the two methods, BMI, and weight were used as predictors of the analyses.

To examine the coherence between the methods to classify different levels of fiber intake, we divided intakes in tertiles from the dietary record and the application respectively. Groups were defined as “low” if the estimated mean fiber intake was equal to or below the lowest tertile, as “moderate” if between the lowest and highest tertile, and as “high” if equal to or higher than the highest tertile.

To identify the most common food groups contributing to the total dietary fiber intake, we summarized dietary fiber quantities from each food group for the whole study population and divided them by the total mean dietary fiber intake of the group. Food items from the dietary record containing less than 0.1 g were excluded from the fiber calculations. The most common food groups are presented as a percentage of the total fiber intake of the group. To evaluate the most common sources of soluble and insoluble fiber in the diet, data from the application were used. The largest food groups of soluble and insoluble fiber intake were calculated and are presented in the same way as the total fiber intake. For comparison of fiber sources between the two methods, each food item contributing to fiber intake was classified into the correct food group for individual data of both methods. The total fiber intake from each food group was then divided by the total fiber intake and presented as a percentage of total fiber intake in individual data.

To estimate if reported energy intakes from the dietary record were reasonable and to identify possible under-reporting, the estimated Resting Energy Expenditure (REE) was calculated with Henry’s [17] for women age 30–60 years which takes both weight and height into consideration. Estimated Physical Activity Level (PAL) was estimated from the Food Intake Level (FIL) calculated as the reported energy intake divided by the estimated REE. Energy stability is required for the assessment of an acceptable FIL over time for survival, given the corresponding PAL. PAL-values below 1.2 were considered as an indication of underestimation of energy intake as no participant was assumed to be confined to bed. Eight dietary record registrations were considered as potentially under-reported due to low estimated PAL values. A sensitivity analysis was done to test if the underestimation of the dietary record affected the result of the correlation.

We simulated a power calculation with a sample size of 20, 30, and 40 participants (Appendix A). In the simulation, 30 participants were required to reach 80 percent power at the alpha level of 0.05 to find a correlation coefficient of 0.48 between the two methods. Several previous studies validating two dietary assessment methods have found a correlation coefficient higher than 0.5 for fiber intake obtained by different dietary assessment methods [18,19,20,21,22,23].

In a subsequent data-driven analysis investigating the impact of specific data points on regression parameter estimates and *p*-values an iterative procedure was applied in which at each step data points were excluded if they deviated more than three standard deviations from the regression line, had high leverage (>3 p/n, p = number of parameters in the regression model; n = number of individuals in the regression model) or had a large Cook’s distance (>median of F (p, n − p), F = F-distribution). The procedure was terminated when no data point met any of these criteria. Twenty-seven women completed both the three-day application registration and three-day dietary record. One woman was excluded from all analyses due to a high deviation (>20 g) of mean fiber intake between the two measurement methods. An investigation concluded that this deviation was probably caused by a significant change in diet composition and not necessarily primarily by the properties of the measurement methods. The iterative procedure—described in the introduction to this section—excluded two participants in sensitivity analyses due to high leverage and one participant due to large Cook’s distance.

A significance threshold for *p*-values of 0.05 was consistently applied. All statistical analyses were performed in R version 4.0.0 and SPSS Statistics version 26.

## 3. Results

### 3.1. Population Characteristics and Energy and Nutrient Intake

Table 1 presents background characteristics and energy- and nutrient intakes. Participants (*n* = 26) had a median age of 57.5 years. The median age amongst the non-participants and those not completing both methods (*n* = 98) was slightly lower, 55.0 years (range = 35–85). The group had a mean PAL value of 1.3 (SD ± 0.26; range = 0.9–1.9). There were no statistically significant differences in mean daily intake of energy- and nutrients or energy distribution between those who started with the application or those who started with the dietary record (Table 1).

Dietary fiber intakes in the two randomized groups and the differences between the two assessment methods are presented in Table 2. The median total fiber intake from both methods was 15.9 g (range = 11.8–35.3). The median difference between the fiber intakes from the dietary record and the application was 2.4 g (95% CI = −0.1–4.2). There was no statistically significant difference in the mean- or difference of fiber intakes between those who started with the application or those who started with the dietary record (Table 2).

### 3.2. Validity of the Mobile Phone Application

#### 3.2.1. Bland-Altman Analyses

The Bland-Altman plot illustrating the agreement between the measurements in the application and the dietary record is shown in Figure 3. The absolute mean difference between the methods was −2.2 g (95% CI = −4.2–(−0.1)). The slope parameter in the regression analysis was not statistically significantly non-zero (*p* = 0.82).

#### 3.2.2. Coherence between the Mobile Phone Application and the Dietary Record and Effects of Demographic Characteristics

We found a statistically significant correlation between the dietary record and application for total fiber intake (rho = 0.65; 95% CI = 0.35–0.83; *p* < 0.001). The correlation plot (Figure 4a) shows a positive linear correlation and that 40% (rho^2^ = 0.422) of the variation in the application can be explained by the variation in the dietary record. The sensitivity analysis—excluding estimated under-reporters—resulted in a slightly higher estimated correlation coefficient between application- and dietary record measurements (rho *=* 0.68; 95% CI = 0.32–0.87; *p* = 0.002). Finally, the iterative procedure—excluding all outliers—once again did not substantially alter the correlation estimates (Appendix A; rho = 0.59; 95% CI = 0.24–0.81; *p* = 0.003). The regression slope estimates using all data, excluding potential under-reporters and after termination of the iterative procedure were in turn: β = 0.67 (95% CI = 0.34–1.01; *p* < 0.001), β = 0.68 (95% CI = 0.30–1.07; *p* = 0.002), and β = 1.04 (95% CI = 0.40–1.68; *p* = 0.003).

There was a non-statistically significant positive linear association between the deviation (in terms of percentage of mean intake between the two measurement methods) and age (Figure 4b; β = 0.65; CI = −0.18–1.47; *p* = 0.12). There was no significant association between the deviation and mean fiber intake of the two methods (β = 1.09; 95% CI = −0.93–3.10; *p* = 0.28), or BMI (β = −0.97; 95% CI = −5.13–3.19; *p* = 0.63), or weight (β = 0.35; 95% CI = −1.02–1.71; *p* = 0.60), results are shown in Appendix A, respectively.

#### 3.2.3. Ability to Classify Fiber Intake

Table 3 shows the classification of dietary fiber intake into low-, moderate- and high intake. Fourteen women (54%) were classified in the same tertile with both methods, eleven women (42%) were classified in the adjacent tertile and one woman (4%) was classified in the opposite tertile.

### 3.3. Sources of Dietary Fiber

The dietary record and the application had the same top three food groups contributing most to the total intake of fiber (Table 4). “Vegetables/roots/potatoes” was the largest source, “bread/crackers” was the second-largest source, and “fruits/berries” was the third-largest source of total fiber intake in both assessment methods. “Other” was only available in the dietary record and contributed to three percent of total fiber intake. “Snacks” contributed the least to total fiber intake, representing one percent in both methods. “Vegetables/roots/potatoes” contributed most to soluble fiber intake and “bread/crackers” contributed most to insoluble fiber intake (Table 5).

### 3.4. Participants’ Experiences

The experience of using the two methods varied widely among the participants. Some preferred the application (38%) and others favored the dietary record (27%). Several participants had a neutral or no opinion about the two assessment methods (35%). There was no difference in age between users who preferred the application and those who preferred the dietary record. The application was considered convenient to have available at all times and thus could be easily used to register foods in close connection with the food intake. The disadvantages of the application were considered to be difficulties in estimating the intake according to the measurements in the application and which foods that contained fiber thus would be registered. The dietary record was considered easy to use because all foods were recorded and the amounts of foods were precise owing to the use of a food scale or the portion guide. However, some participants found it tiring to register all foods in detail. Participants’ self-reported special diets or allergies did not systematically seem to affect dietary fiber intakes or the difference of fiber intakes between the two methods. Four participants registered over a Swedish holiday although major differences between the methods were only seen in three of these participants’ fiber intakes. Moreover, one of these three participants was excluded as an outlier in the sensitivity analysis. Reported circumstances due to the pandemic situation were, for instance, receiving help with grocery shopping and working from home. The majority did not think these circumstances had affected their dietary intake during the periods of registrations and these factors did not vary between the two dietary assessment periods.

## 4. Discussion

In a crossover study of 26 individuals, we compared dietary fiber intake registered with a mobile phone application and a three-day dietary record. We found that the use of the application led to an underestimation of the fiber intake by two grams per day compared to the dietary record. The absolute underestimation was independent of total fiber intake and the relative underestimation decreased with increasing age. The most important sources of total and soluble dietary fiber were vegetables and roots, and the largest sources of insoluble fiber were bread and crackers.

The use of the application led to an underestimation of the dietary fiber intake compared to the dietary record. We have not found any other study comparable to ours validating fiber intake measurements by using a mobile phone application against a dietary record. However, in validation studies comparing food frequency questionnaires and dietary records, underestimations of 2–3 g per day have been found for measurements with food frequency questionnaires [23,24]. The underestimation could partly be explained by the absence of some food items in the application, items that contribute to fiber intake. Another explanation is that the participants lacked knowledge of what food items they should register when they were eating prepared dishes. Our estimated correlation coefficient of rho = 0.65 was in agreement with estimates from other studies validating fiber intakes by using other types of dietary assessment methods against dietary records; correlation coefficients ranging from 0.43 to 0.72 have been identified [18,19,20,22,23,24,25,26,27]. In some studies, the correlation coefficients increased when adjusting for energy intakes [18,24,26,27]. Energy adjustment was not possible in our study since the application did not target the totality of the diet and thereby lacks the necessary information.

The absolute difference between the results from the two methods was independent of the participants‘ total fiber intake and the relative difference decreased with increasing age. Vuholm and coworkers [23] also found that the underestimation in results from subjects using a food frequency questionnaire compared to a dietary record was independent of the average fiber intake. Other similar studies have reported larger variance at lower fiber intakes than higher [20] and the opposite, increasing difference with increasing intakes [24,28]. The ability of the application to classify fiber intakes in the right tertile suggests that the use of the application will be more useful in ranking an individual’s fiber intake than identifying the individual’s absolute intake. Categorical intake levels are often sufficient to answer nutritional research hypotheses [29]. The classification was in line with previous findings, where 81–90% of the fiber intakes have been categorized in the same or adjacent quartile or quintile [18,20,22]. Regarding our findings of decreased underestimation with increasing age, a Brazilian study found a higher correlation coefficient for energy-adjusted fiber intakes between a food frequency questionnaire and 24-h recalls in elderly participants compared to younger [30]. A higher percentage of elderly was also classified in the same or adjacent quartile for the methods compared to younger participants. Compared to younger participants, older people may register dietary intake more precisely or their dietary habits are more constant between registration periods.

We found that vegetables and roots contributed most to total fiber intake and bread and crackers were the second-largest sources, but this finding differs from two previous investigations of Swedish women [31,32]. In those investigations, the largest source of fiber was cereals respectively bread, and then vegetables [31,32]. The reported fiber intakes in those investigations were 14 g respectively 19 g per day compared to ours of 16 g per day. These intakes are lower than the Swedish recommendations of 25–35 g of dietary fiber per day [10]. Since we investigated only a small sample of women, our results may not be representable for the entire Swedish population.

The strengths of this study include the random selection of study participants, the crossover design where participants used both instruments in random order. Of those participants who started the registration, the dropout rate was low. We included Friday in the three days of registration to represent a day of the weekend because eating patterns vary on weekdays and weekends [6]. To avoid the different methods consistently affecting the other, we chose a randomized crossover design. A design of participants using the test- and reference method during the same period might be preferable in a validation study because then the only differences are the methods’ ability to measure dietary intake. In our study, using that approach would have meant that participants weighed, measured, or used the portion guide to estimate food intake for the dietary record and then in close connection registered with the application and converted the amounts to relevant units. However, the participants’ true fiber intake could have differed between the registration periods.

A limitation in all validation studies is whether the reference method reflects the true dietary intake or not. Using weighed dietary records places a high burden on respondents and that can affect the amount and type of food eaten, nevertheless, the use of these records is still one of the best available tools, and measuring errors are probably partly uncorrelated to those of the application [5]. Three days of registration of each assessment method are not sufficient to capture individual habitual fiber intake [33], but we chose that number of days of registration to achieve higher compliance with the intervention. We used no objective biomarkers for dietary fiber because to our knowledge, at this time point, there is no optimal biomarker for total fiber intake [1,34,35]. Biomarkers likewise have limitations [36], but the advantage of using them in this study would have been to control reported dietary intakes—if differences between the two methods were a result of misreporting of foods or true different fiber intakes during the periods.

We tried to keep the washout period as close to two weeks as possible and intended to send the instructions for the second method after an interval that would ensure that they would not arrive until at least two weeks after the first conducted method. A long period between registrations, might lead to seasonal variations affecting food intake and thus also fiber intake. This could have influenced some of the registrations but apart from two participants with washouts of 8 and 11 weeks respectively, the range was only one to five weeks.

The results may not be relevant for men due to known variations between genders in food intake as well digital competence [24,33]. Moreover, there may be cultural factors concerning food intake and digital competence in all genders. We wanted a homogenous group and only included women in the study. Women have previously had a higher participation rate in dietary studies than men [20,22,32], indicating that women more willingly participate in these studies. Although higher correlation coefficients have been reported for men compared to women [24]. Our study population might also have higher technical knowledge than corresponding women in the average population. Some of the women declined participation due to technical difficulties. We are not testing a product; we believe the findings can be generalized to other similar applications.

When interpreting data obtained by using the application, the real intake should be considered higher than the intake registered. Furthermore, the application can be used regardless of the size of the fiber intake. However, age must be considered when interpreting the difference between real- and estimated intake with the application. In addition, if dietary fiber intake should be investigated—food items of vegetables, roots, bread, and crackers should be prioritized. Our results will be used to further develop the application and we will add about 30 complimentary food items to decrease the underestimation and ease the registration for users. Examples of these foods are oat milk, orange juice, cookies, buns, physalis, jam, sweet potato, artichoke, mixed salad, pizza crust, and bread for sausages and hamburgers. The application may be a more convenient and user-friendly method as it requires less effort, but there may be individual variations.

## 5. Conclusions

We can assume that the true dietary fiber intake at a group level is at least two grams higher than the measured fiber intake recorded by use of the application, but the absolute underestimation does not vary with the size of the measured total intake. Vegetables and roots contributed most to total and soluble fiber intake, bread and crackers contributed most to insoluble fiber intake. The application may be considered as a useful and easy-to-use method to measure dietary fiber intake. Many, but not all, have a mobile phone and experience in using applications.

## Figures and Tables

**Figure 1 nutrients-13-02133-f001:**
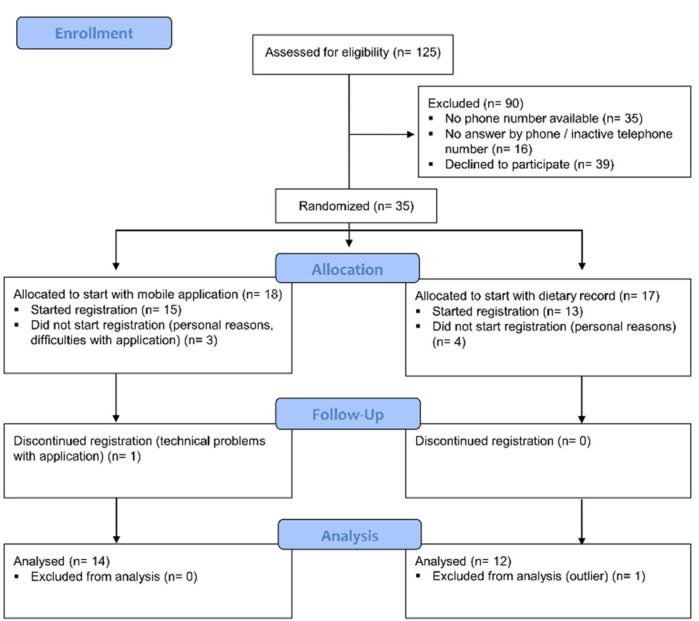
Flowchart of the study selection procedure.

**Figure 2 nutrients-13-02133-f002:**
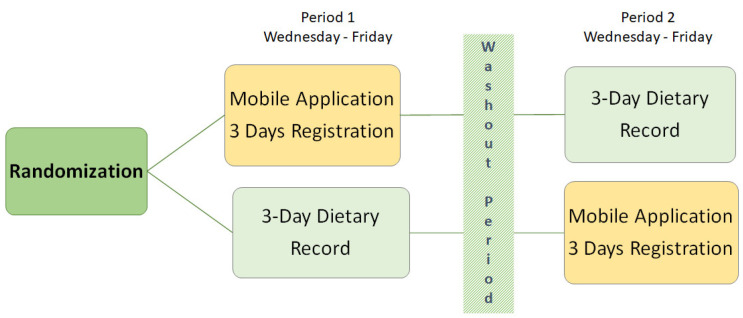
Flow chart of the study design.

**Figure 3 nutrients-13-02133-f003:**
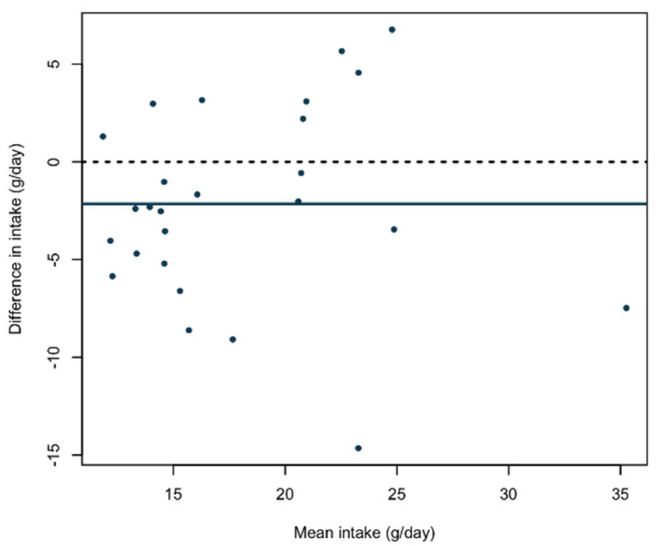
Bland-Altman plot showing the difference in absolute mean fiber intake between the mobile phone application and dietary record as a function of the mean fiber intake of the methods (*n* = 26). The solid line represents the mean absolute difference of fiber intake between the methods. The dashed line at zero is included for reference.

**Figure 4 nutrients-13-02133-f004:**
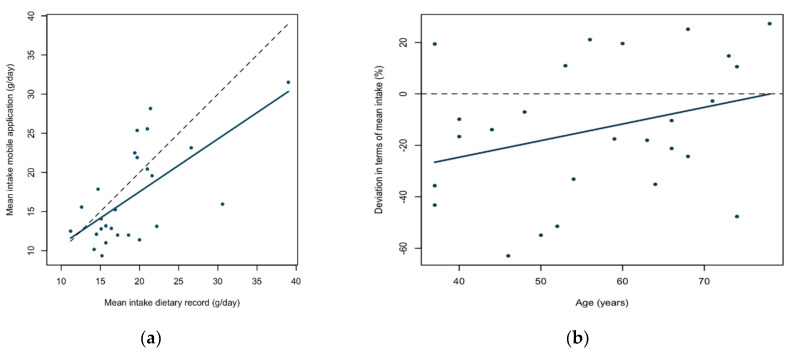
(**a**) Pearson correlation plot of correlation between mean fiber intake (g) measured by the mobile phone application and the dietary record (*n* = 26). The dashed line presents a unity line (x = y). (**b**) Regression analysis including the deviation in terms of percentage of mean fiber intake as measured by the mobile phone application and the dietary record and age (*n* = 26). The dashed line at zero is included for reference.

**Table 1 nutrients-13-02133-t001:** Baseline characteristics, daily energy- and nutrient intake, and energy distribution (E %) reported from the dietary records presented for all women and separately for the two randomized groups.

	All Women (*n* = 26)			Started with Mobile Phone Application (*n* = 14) ^2^			Started with Dietary Record (*n* = 12) ^2^		
	Median	IQR ^3^	Range	Median	IQR ^3^	Range	Median	IQR ^3^	Range
Age (years)	57.5	23.0	37–78	58.0	27.0	37.0–73.0	56.5	23.0	40.0–78.0
Height (cm)	168.5	6.0	160.0–183.0	170.0	8.0	162.0–83.0	168.5	7.0	160.0–173.0
Weight (kg)	69.8	11.3	55.0–86.0	68.9	11.5	56.0–86.0	71.0	11.6	55.0–83.0
BMI (kg/m^2^)	24.1	4.4	20.4–30.5	23.7	3.0	21.0–30.5	25.9	5.0	20.4–30.1
Energy (kcal)	1817	576	1305–2636	1977	600	1471–2636	1733	568	1305–2473
Protein (g)	72	25	44–134	73	24	47–134	70	30	44–97
Protein (E%)	17	3	8–26	17	5	8–26	17	3	12–23
Carbohydrates (g) ^1^	186	65	110–242	191	70	110–242	174	63	130–228
Carbohydrates (E%) ^1^	43	11	29–58	42	7	29–58	44	12	34–55
Fats (g)	86	43	47–140	93	42	47–140	73	36	48–129
Fats (E%)	40	10	26–50	42	6	26–49	38	12	32–50
Saturated fats (E%)	15	4	9–23	14	5	9–20	16	3	11–23

^1^ Including dietary fiber and alcohol. ^2^ Mann-Whitney U test found no statistically significant difference in any variable between the two groups. *p*-values of ≤0.05 were considered statistically significant for all statistical analyzes. ^3^ Interquartile range (IQR) representing values between the first and the third quartiles.

**Table 2 nutrients-13-02133-t002:** Dietary fiber intakes and the difference between the assessment methods presented for all women and separately for the two randomized groups.

	All Women (*n* = 26)			Started with Mobile Application (*n* = 14) ^2^			Started with Dietary Record (*n* = 12) ^2^		
	Median	IQR ^3^	Range	Median	IQR ^3^	Range	Median	IQR ^3^	Range
Fiber intake application (g)	15	10	9–32	17	12	11–32	13	8	9–28
Fiber intake dietary record (g)	18	6	11–39	18	6	13–39	18	6	11–31
Mean fiber intake both methods (g)	16	7	12–35	16	8	13–35	16	8	12–25
Difference (g) ^1^	2	8	−7–15	2	8	−6–8	2	9	−7–15

^1^ Difference in dietary fiber intake between the dietary record and the mobile phone application (dietary record − mobile phone application). ^2^ Mann-Whitney U test found no statistically significant difference in any variable between the two groups. *p*-values of ≤0.05 were considered statistically significant for all statistical analyzes. ^3^ Interquartile range (IQR) representing values between the first and the third quartiles.

**Table 3 nutrients-13-02133-t003:** Classification of dietary fiber intake into a low, moderate, or high intake of total fiber intake registered by the mobile phone application and the dietary record.

	Mobile Phone Application			
Dietary Record	Low, *n* (%)	Moderate, *n* (%)	High, *n* (%)	Total
Low, *n* (%)	5 (19)	3 (12)	0 (0)	8 (31)
Moderate, *n* (%)	3 (12)	3 (12)	3 (12)	9 (35)
High, *n* (%)	1 (4)	2 (8)	6 (23)	9 (35)
Total	9 (35)	8 (31)	9 (35)	26 (100)

The percentages were rounded to the nearest integer.

**Table 4 nutrients-13-02133-t004:** The three food groups that contributed the most to total dietary fiber intake from registrations in the mobile phone application and the dietary records (*n* = 26).

Top 3	Food Groups from the Mobile Phone Application	Food Groups from the Dietary Record
1	Vegetables/roots/potatoes (30%)	Vegetables/roots/potatoes (27%)
2	Bread/crackers (25%)	Bread/crackers (26%)
3	Fruits/berries (15%)	Fruits/berries (17%)

**Table 5 nutrients-13-02133-t005:** Top three food groups that contributed the most to soluble- and insoluble fiber intake from registrations in the mobile phone application (*n* = 26).

Top 3	Food Groups of Soluble Fiber	Food Groups of Insoluble Fiber
1	Vegetables/roots/potatoes (36%)	Bread/crackers (29%)
2	Fruits/berries (21%)	Vegetables/roots/potatoes (26%)
3	Bread/crackers (16%)	Fruits/berries (15%)

## Data Availability

The data presented in this study are available on request to the corresponding author. The data are not publicly available due to analyses that are ongoing for future publications. Data will be offered if requested.

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
