# Peer review of "Development and Validation of a Mobile Phone Application Developed for Measuring Dietary Fiber Intake"

_nutrients, 2021, doi:10.3390/nu13072133_

Round 1
Reviewer 1 Report
The use of a mobile phone application for measuring the intake of dietary fiber can be a very simple and convenient tool supporting the work of a dietitian with a patient.
line 72-73 Why was so small a randomly selected group of women (n = 125) was identified from the Swedish Population. What were the criteria that the authors did not invite 1000 women to the study, which would give about 200-250 participants and then the research group would be sufficient?
line 127 - The authors indicate that the application contained 51
dishes from recipes, were they the most eaten dishes in Sweden, regional dishes, ready-made dishes available in supermarkets?
why only 51? What about the dishes that the women participating in the study prepared at home according to their own recipes? was there any dishes other than typically Swedish? fast food?
due to the very small group of people participating in the study, I would suggest rewriting the title and adding information that it is a preliminary / pilot study.
no binding conclusions can be drawn on such a small research group.
Author Response
Point 1: The use of a mobile phone application for measuring the intake of dietary fiber can be a very simple and convenient tool supporting the work of a dietitian with a patient.
Response 1: Thank you.
Point 2: line 72-73 Why was so small a randomly selected group of women (n = 125) was identified from the Swedish Population. What were the criteria that the authors did not invite 1000 women to the study, which would give about 200-250 participants and then the research group would be sufficient?
Response 2: Thank you for this comment. We have no reason to believe that a larger study would provide a change in the interpretation of the results (findings). According to the power calculation, we needed a sample size of 30 participants to get a correlation of 0.48 between the measurements of the two methods.
Point 3: line 127 - The authors indicate that the application contained 51 dishes from recipes, were they the most eaten dishes in Sweden, regional dishes, ready-made dishes available in supermarkets?
Response 3: We agree with the reviewer that this information is missing. We have now added the text “The recipes are registered in portions and are easy-to-prepare, typically prevalent in the Swedish diet (e.g., sandwiches, porridges, stews, soups), and can be tolerated during pelvic radiotherapy. The recipes were added to the application to ease the registration for participants in the FIDURA-study.” on line 133.
Point 4: why only 51? What about the dishes that the women participating in the study prepared at home according to their own recipes? was there any dishes other than typically Swedish? fast food?
Response 4: The foods and dishes that were included in the mobile phone application were judged to be most important for a healthy microbiota and at the same time common in the Swedish diet. We did not add more recipes to the application because we noticed that the patients in our preparatory study of FIDURA often did not use them. As we answered in Response 3, the information has been expanded to “The recipes are registered in portions and are easy-to-prepare, typically prevalent in the Swedish diet (e.g., sandwiches, porridges, stews, soups), and could be tolerated during pelvic radiotherapy. The recipes were added to the application to ease the registration for participants in the FIDURA-study.” on line 133.”
Point 5: due to the very small group of people participating in the study, I would suggest rewriting the title and adding information that it is a preliminary / pilot study.
no binding conclusions can be drawn on such a small research group
Response 5: We respectfully disagree. We believe that the interpretation of our results is reasonable. Please note that we are not testing a product, we believe the findings can be generalized to other similar mobile phone applications.
Reviewer 2 Report
The research report was very well prepared. The authors properly described the purpose of the study and the introduction. Very well-chosen research methods and statistical methods deserve to be emphasized. The results are presented clearly. The discussion was properly prepared.
My doubts are only the possibility of broader use of the application than the studied population due to differences resulting from eating habits and product availability. However, methodically, the work is very well prepared.
Author Response
Point 1: The research report was very well prepared. The authors properly described the purpose of the study and the introduction. Very well-chosen research methods and statistical methods deserve to be emphasized. The results are presented clearly. The discussion was properly prepared.
Response 1: Thank you.
Point 2: My doubts are only the possibility of broader use of the application than the studied population due to differences resulting from eating habits and product availability. However, methodically, the work is very well prepared.
Response 2: We agree. To clarify we have modified the text about the generalizability to: “The results may not be relevant for men due to known variations between genders in food intake as well digital competence [24,34]. Moreover, there may be cultural factors concerning food intake and digital competence in all genders. We wanted a homogenous group and only included women in the study. Women have previously had a higher participation rate in dietary studies than men [20,22,32], indicating that women more willingly participate in these studies. Although higher correlation coefficients have been reported for men compared to women [24]. Our study population might also have higher technical knowledge than corresponding women in the average population. Some of the women declined participation due to technical difficulties. We are not testing a product; we believe the findings can be generalized to other similar applications.”
Reviewer 3 Report
I congratulate the authors on a well-written, informative , empirical investigation comparing using a phone app with weighed inventory for assessing fibre intake.
The design is strong (RCT with crossover) and entirely appropriate for the intention of the investigation.
I find it clear and well-written in all sections and the work will undoubtedly be interesting and useful for dietitians, nutritionists, public health works and researchers alike.
A couple of minor points:
P3. In your flow chart you state the n for excluded as 90 but only initially account for 89- I think number 90 is your outlier- can you somehow make it clearer? (a line linking the otulier to the excluded number?)
Table 3 total adds up to 101 not 100?
Author Response
Point 1: I congratulate the authors on a well-written, informative, empirical investigation comparing using a phone app with weighed inventory for assessing fibre intake.
The design is strong (RCT with crossover) and entirely appropriate for the intention of the investigation.
I find it clear and well-written in all sections and the work will undoubtedly be interesting and useful for dietitians, nutritionists, public health works and researchers alike.
Response 1: Thank you very much.
Point 2: P3. In your flow chart you state the n for excluded as 90 but only initially account for 89- I think number 90 is your outlier- can you somehow make it clearer? (a line linking the otulier to the excluded number?)
Response 2: Thank you for seeing this error. One participant with an inactive telephone number has been missed in the excluded participants and we will change the category “No answer by phone (n = 15)” to “No answer by phone / inactive telephone number (n= 16)” in the Flowchart.
Point 3: Table 3 total adds up to 101 not 100?
Response 3: Thank you for pointing out that a footnote is missing explaining this fact. The percentages in Table 3 are rounded to the nearest integer, which makes the number 101. We have now added a footnote as follows “The percentages were rounded to the nearest integer”.